# THE POWER OF CHOICES IN DECISION TREE LEARNING

## ABSTRACT

We propose a simple and natural generalization of standard and empirically successful decision tree learning algorithms such as ID3, C4.5, and CART. These classic algorithms, which have been central to machine learning for decades, are greedy in nature: they grow a decision tree by iteratively splitting on the "best" attribute. We augment these algorithms with an additional greediness parameter $k$ and our resulting algorithm, Top-$k$, considers the $k$ best attributes as possible splits instead of just the single best attribute.

We demonstrate, theoretically and empirically, the power of this simple generalization. We first prove a sharp *greediness hierarchy theorem* showing that for every $k \in \mathbb{N}$, Top-$(k+1)$ can be much more powerful than Top-$k$: there are data distributions for which the former achieves accuracy $1 - \varepsilon$, whereas the latter only achieves accuracy $\frac{1}{2} + \varepsilon$. We then show, through extensive experiments, that Top-$k$ compares favorably with the two main approaches to decision tree learning: classic greedy algorithms and more recent "optimal decision tree" algorithms. On one hand, Top-$k$ consistently enjoys significant accuracy gains over the greedy algorithms across a wide range of benchmarks, at the cost of only a mild training slowdown. On the other hand, Top-$k$ is markedly more scalable than optimal decision tree algorithms, and is able to handle dataset and feature set sizes that remain beyond the reach of these algorithms.

Taken together, our results highlight the potential practical impact of the power of choices in decision tree learning.

## 1 INTRODUCTION

Decision trees are a fundamental workhorse in machine learning. Their logical and hierarchical structure makes them easy to understand and their predictions easy to explain. Decision trees are therefore the most canonical example of an interpretable model: in his influential survey (Breiman, 2001b), Breiman writes "On interpretability, trees rate an A+"; much more recently, the survey Rudin et al. (2022) lists decision tree optimization as the very first of 10 grand challenges for the field of interpretable machine learning. Decision trees are also at the heart of modern ensemble methods such as random forests (Breiman, 2001a) and XGBoost (Chen & Guestrin, 2016), which achieve state-of-the-art accuracy for a wide range of tasks.

Greedy algorithms such as ID3 (Quinlan, 1986), C4.5 (Quinlan, 1993), and CART (Breiman et al., 1984) have long been the standard approach to decision tree learning. These algorithms build a decision tree from labeled data in a top-down manner, growing the tree by iteratively splitting on the "best" attribute as measured with respect to a certain potential function (e.g. information gain). Owing to their simplicity, these algorithms are highly efficient and scale gracefully to handle massive datasets and feature set sizes, and they continue to be widely employed in practice and enjoy significant empirical success. For the same reasons, these algorithms are also part of the standard curriculum in introductory machine learning and data science courses.

The trees produced by these greedy algorithms are often reasonably accurate, but can nevertheless be suboptimal. There has therefore been a separate line of work, which we overview in Section 2, on algorithms that optimize for accuracy, and in fact, seek to produce *optimally* accurate decision trees. These algorithms employ a variety of optimization techniques (including dynamic programming, integer programming, and SAT solvers) and are completely different from the simple greedy algorithms discussed above. Since the problem of finding an optimal decision tree has long been known to be NP-hard (Hyafil & Rivest, 1976), *any* algorithm must suffer from the inherent combinatorial

explosion when the instance size becomes sufficiently large (unless P=NP). Therefore, while this line of work has made great strides in improving the scalability of algorithms for optimal decision trees, dataset and feature set sizes in the high hundreds understandably remain out of reach.

This state of affairs raises a natural question:

> *Can we design decision tree learning algorithms that improve significantly on the accuracy of classic greedy algorithms and yet inherit their simplicity and scalability?*

In this work, we propose a new approach and make the case that it provides a strong affirmative answer to the question above. We further show that it opens up several new avenues for exploration in both the theory and practice of decision tree learning.

## 1.1 OUR CONTRIBUTIONS

### 1.1.1 TOP-$k$: A SIMPLE AND EFFECTIVE GENERALIZATION OF CLASSIC GREEDY DECISION TREE ALGORITHMS

We introduce an easily interpretable greediness parameter to the class of all greedy decision tree algorithms, a broad class that encompasses ID3, C4.5, and CART. This parameter, $k$, represents the number of features that the algorithm considers as candidate splits at each step. Setting $k = 1$ recovers the fully greedy classical approaches, and increasing $k$ allows the practitioner to produce more accurate trees at the cost of only a mild training slowdown. The focus of our work is on the regime where $k$ is a small constant—preserving the efficiency and scalability of greedy algorithms is a primary objective of our work—although we mention here that by setting $k$ to be the dimension $d$, our algorithm produces an optimal tree. Our overall framework can thus be viewed as interpolating between greedy algorithms at one extreme and "optimal decision tree" algorithms at the other, precisely the two main and previously disparate approaches to decision tree learning discussed above.

We will now describe our framework. A *feature scoring function* $\mathcal{H}$ takes as input a dataset over $d$ binary features and a specific feature $i \in [d]$, and returns a number quantifying the "desirability" of this feature as the root of the tree. The greedy algorithm corresponding to $\mathcal{H}$ selects as the root of the tree it builds the feature that has the largest score under $\mathcal{H}$; our generalization will instead consider the $k$ features with the $k$ highest scores.

**Definition 1** (Feature scoring function). *A feature scoring function $\mathcal{H}$ takes as input a labeled dataset $S$ over a $d$-dimensional feature space, a feature $i \in [d]$, and returns a score $\nu_i \in [0, 1]$.*

See Section 3.1 for a discussion of the feature scoring functions that correspond to standard greedy algorithms ID3, C4.5, and CART. Pseudocode for Top-$k$ is provided in Figure 1. We note that from the perspective of interpretability, the trained model looks the same regardless of what $k$ is. During training, the algorithm considers more splits, but only one split is eventually used at each node.

### 1.1.2 THEORETICAL RESULTS ON THE POWER OF TOP-$k$

The search space of Top-$(k + 1)$ is larger than that of Top-$k$, and therefore its training accuracy is certainly at least as high. The first question we consider is: Is the test accuracy of Top-$(k + 1)$ only marginally better than that of Top-$k$, or are there examples of data distributions for which even a single additional choice provably leads to huge gains in test accuracy? Our first main theoretical result is a sharp *greediness hierarchy theorem*, showing that this parameter can have dramatic impacts on accuracy, thereby illustrating its power:

**Theorem 1** (Greediness hierarchy theorem). *For every $\varepsilon > 0$, $k, h \in \mathbb{N}$, there is a data distribution on which Top-$(k + 1)$ achieves at least $1 - \varepsilon$ accuracy with a depth budget of $h$, but Top-$k$ achieves at most $\frac{1}{2} + \varepsilon$ accuracy with a depth budget of $h$.*

Theorem 1 is a special case of a more general result that we show: for all $k < K$, there are data distributions on which Top-$K$ achieves maximal accuracy gains over Top-$k$, even if Top-$k$ is allowed a larger depth budget:

**Theorem 2** (Generalization of Theorem 1). *For every $\varepsilon > 0$, $k, K, h \in \mathbb{N}$ where $k < K$, there is a data distribution on which Top-$K$ achieves at least $1 - \varepsilon$ accuracy with a depth budget of $h$, but Top-$k$ achieves at most $\frac{1}{2} + \varepsilon$ accuracy even with a depth budget of $h + (K - k - 1)$.*

Top-$k(\mathcal{H}, S, h)$:

> **Given:** A feature scoring function $\mathcal{H}$, a labeled sample set $S$ over $d$ dimensions, and depth budget $h$.
>
> **Output:** Decision tree of depth $h$ that approximately fits $S$.
>
> 1. If $h = 0$, or if every point in $S$ has the same label, return the constant function with the best accuracy w.r.t. $S$.
>
> 2. Otherwise, let $\mathcal{I} \subseteq [d]$ be the set of $k$ coordinates maximizing $\mathcal{H}(S, i)$.
>
> 3. For each $i \in \mathcal{I}$, let $T_i$ be the tree with
>
> $$\text{Root} = x_i$$
> $$\text{Left subtree} = \text{Top-}k(\mathcal{H}, S_{x_i=0}, h-1)$$
> $$\text{Right subtree} = \text{Top-}k(\mathcal{H}, S_{x_i=1}, h-1),$$
>
> where $S_{x_i=b}$ is the subset of points in $S$ where $x_i = b$.
>
> 4. Return the $T_i$ with maximal accuracy with respect to $S$ among all choices of $i \in \mathcal{I}$.

Figure 1: The Top-$k$ algorithm. It can be instantiated with any feature scoring function $\mathcal{H}$, and when $k = 1$, recovers standard greedy algorithms such as ID3, C4.5, and CART.

The proof of Theorem 2 is simple and cleanly highlights the theoretical power of choices. One downside, though, is that it is based on data distributions that are admittedly somewhat unnatural: the labeling function has embedded within it a function that is the XOR of certain features, and real-world datasets are unlikely to exhibit such adversarial structure. To address this, we further prove that the power of choices is evident even for *monotone* data distributions.

**Theorem 3** (Greediness hierarchy theorem for monotone data distributions). *For every $\varepsilon > 0$, depth budget $h$, $K$ between $\tilde{\Omega}(h)$ and $\tilde{O}(h^2)$ and $k \le K - h$, there is a monotone data distribution on which Top-K achieves at least $1 - \varepsilon$ accuracy with a depth budget of $h$, but Top-k achieves at most $\frac{1}{2} + \varepsilon$ accuracy with a depth budget of $h$.*

Many real-world data distributions are monotone in nature, and relatedly, they are a common assumption and the subject of intensive study in learning theory. Most relevant to this paper, recent theoretical work has identified monotone data distributions as a broad and natural class for which classical greedy decision tree algorithms (i.e. Top-1) provably succeed (Blanc et al., 2020b;a). Theorem 3 shows that even within this class, increasing the greediness parameter can lead to dramatic gains in accuracy. Compared to Theorem 2, the proof of Theorem 3 is more technical and involves the use of concepts from the Fourier analysis of boolean functions (O'Donnell, 2014).

We note that a weaker version of Theorem 3 is implicit in prior work: Combining (Blanc et al., 2020b, Theorem 7b) and (Blanc et al., 2021b, Theorem 2) yields the special case of Theorem 3 where $K = O(h^2)$ and $k = 1$. Theorem 3 is a significant strengthening as it allows for $k > 1$ and $K - k$ much smaller.

### 1.1.3 Experimental results on the power of Top-$k$

We provide extensive empirical validation of the effectiveness of Top-$k$ when trained on on real-world datasets, and provide an in-depth comparison with both standard greedy algorithms as well as optimal decision tree algorithms.

We first compare the performance of Top-$k$ for $k = 1, 2, 3, 4$ (Figure 2), and find that increasing $k$ does indeed provide a significant increase in test accuracy—in a number of cases, Top-4 already achieves accuracy that is comparable to the test accuracy attained by MurTree (Demirović et al., 2022), a state-of-the-art optimal decision tree algorithm. We further show, in Figures 3 and 4, that Top-$k$ inherits the efficiency of popular greedy algorithms, and furthermore scales much better than the optimal decision tree algorithms MurTree and GOSDT (Lin et al., 2020).

Taken as a whole, our experiments demonstrate that Top-$k$ provides a useful middle ground between greedy and optimal decision tree algorithms: It is significantly more accurate than greedy algorithms, but still fast enough to be practical on reasonably large data sets. See Section 5 for an in-depth discussion of our experiments. Finally, we emphasis the benefits afforded by the simplicity of Top-$k$. Standard greedy algorithms (i.e. Top-1) are widely employed and easily accessible. Introducing the parameter $k$ requires modifying only a tiny amount of source code and gives the practitioner a new lever to control. Our experiments and theoretical results demonstrate the utility of this simple lever.

## 2   RELATED WORK

**Provable guarantees and limitations of greedy decision tree algorithms.**   A long and fruitful line of work seeks to develop a rigorous understanding of the performances of greedy decision tree learning algorithms such as ID3, C4.5, and CART, and to place their empirical success on firm theoretical footing (Kearns & Mansour, 1996; Kearns, 1996; Dietterich et al., 1996; Brutzkus et al., 2019; 2020; Blanc et al., 2020b;a; 2021a). These works identify feature and distributional assumptions under which these algorithms provably succeed; they also highlight the *limitations* of these algorithms by pointing out settings on which they provably fail. Our work complements this line of work by showing, theoretically and empirically, how these algorithms can be further improved with a simple new parameter while preserving their efficiency and scalability.

**The work of Blanc et al. (2021b).**   Recent work of Blanc et al. also highlights the power of choices in decision tree learning. However, they operate within a stylized theoretical setting. First, they consider a specific scoring function that is based on a notion of influence of features, and crucially, computing these scores requires *query access* to the target function (rather than from random labeled samples as is the case in practice). Furthermore, their results only hold with respect to the uniform distribution. These are strong assumptions that limit the practical relevance of their results. In contrast, a primary focus of this work is to be closely aligned with practice, and in particular, our framework captures and generalizes the standard greedy algorithms used in practice.

**Optimal decision trees.**   Motivated in part by the surge of interest in interpretable machine learning and the highly interpretable nature of decision trees, there have been numerous works on learning *optimal* decision trees (Bertsimas & Dunn, 2017; Verwer & Zhang, 2017; 2019; Aghaei et al., 2019; Zhu et al., 2020; Verhaeghe et al., 2020; Narodytska et al., 2018; Avellaneda, 2020; Janota & Morgado, 2020; Nijssen & Fromont, 2007; 2010; Hu et al., 2019; Lin et al., 2020; Demirović et al., 2022). As mentioned in the introduction, this is an NP-complete problem (Hyafil & Rivest, 1976)— indeed, it is NP-hard to find even an approximately optimal decision tree (Sieling, 2008; Adler & Heeringa, 2008; Alekhnovich et al., 2009). Due to the fundamental intractability of this problem, even highly optimized versions of algorithms are unlikely to match the scalability of standard greedy algorithms. That said, these works implement a variety of optimizations that allow them to build optimal decision trees for many real world data sets when the data set and feature sizes are in the hundreds and the desired depth is small ($\approx 5$ or less).

Finally, another related line of work is that of *soft* decision trees (Irsoy et al., 2012; Tanno et al., 2019). These works use gradient-based methods to learn soft splits at each internal node. We believe that one key advantage of our work over these soft trees is in interpretability. With Top-k, since the splits are hard (and not soft), to understand the classification of a test point, it is sufficient to look at only one root-to-leaf path, as opposed to a weighted combination across many.

## 3   THE TOP-$k$ ALGORITHM

### 3.1   BACKGROUND AND CONTEXT: IMPURITY-BASED ALGORITHMS

Greedy decision tree learning algorithms like ID3, C4.5 and CART are all instantiations of Top-$k$ in Figure 1 with $k = 1$ and an appropriate choice of the feature-scoring function $\mathcal{H}$. Those three algorithms all used *impurity-based heuristics* as their feature-scoring function:

**Definition 2** (Impurity-based heuristic). *An impurity function $\mathcal{G} : [0, 1] \to [0, 1]$ is a function that is concave, symmetric about $0.5$, and satisfies $\mathcal{G}(0) = \mathcal{G}(1) = 0$ and $\mathcal{G}(0.5) = 1$. A feature-scoring*

*function $\mathcal{H}$ is an impurity-based heuristic, if there is some impurity function $\mathcal{G}$ for which:*

$$\mathcal{H}(S, i) = \mathcal{G}\left(\mathop{\mathbb{E}}_{\boldsymbol{x},\boldsymbol{y}\sim S}[\boldsymbol{y}]\right)$$

$$- \mathop{\Pr}_{\boldsymbol{x},\boldsymbol{y}\sim S}[\boldsymbol{x}_i = 0] \cdot \mathcal{G}\left(\mathop{\mathbb{E}}_{\boldsymbol{x},\boldsymbol{y}\sim S}[\boldsymbol{y} \mid \boldsymbol{x}_i = 0]\right) - \mathop{\Pr}_{\boldsymbol{x},\boldsymbol{y}\sim S}[\boldsymbol{x}_i = 1] \cdot \mathcal{G}\left(\mathop{\mathbb{E}}_{\boldsymbol{x},\boldsymbol{y}\sim S}[\boldsymbol{y} \mid \boldsymbol{x}_i = 1]\right)$$

*where in each of the above, $(\boldsymbol{x}, \boldsymbol{y})$ are a uniformly random point from within $S$.*

Common examples for the impurity function include the binary entropy function, $\mathcal{G}(p) = -p\log_2(p) - (1-p)\log_2(1-p)$ (used by ID3 and C4.5), the Gini index $\mathcal{G}(p) = 4p(1-p)$ (used by CART), and the $\mathcal{G}(p) = 2\sqrt{p(1-p)}$ (proposed and analyzed in Kearns & Mansour (1999)). We refer to the reader to Kearns & Mansour (1999) for a theoretical comparison, and Dietterich et al. (1996) for an experimental comparison, of these impurity-based heuristics.

Our experiments focus on Gini index being the impurity measure, but our theoretical results apply to Top-$k$ instantiated with *any* impurity-based heuristic.

### 3.2 BASIC THEORETICAL PROPERTIES OF THE TOP-$k$ ALGORITHM

**Running time.** The key behavioral aspect in which Top-$k$ differs from greedy algorithms is that it is less greedy when trying to determine which coordinate to query. This naturally increases the running time of Top-$k$, but that increase is fairly mild. Concretely, say that Top-$k$ is run on a dataset $S$ with $n$ points. We can then easily derive the following bound on the running time of Top-$k$, where $\mathcal{H}(S, i)$ is assumed to take $O(n)$ time to evaluate (as it does for all impurity-based heuristics).

**Claim 3.1.** *The running time of Top-$k(\mathcal{H}, S, h)$ is $O((2k)^h \cdot nd)$.*

*Proof.* Let $T_h$ be the number of recursive calls made by Top-$k(\mathcal{H}, S, h)$. Then, we have the simple recurrence relation $T_h = 2kT_{h-1}$, where $T_0 = 1$. Solving this recurrence gives $T_h = (2k)^h$. Each recursive call takes $O(nd)$ time, where the bottleneck is scoring each of the $d$ features. $\square$

We note that any decision tree algorithm, including blazingly fast greedy algorithms such as ID3, C4.5, and CART, has runtime that scales exponentially with the depth, $h$. The size of a depth-$h$ can be $2^h$, and this is of course a lower bound on the runtime as the algorithm needs to output such a tree. In particular, contrasting the running time of Top-$k$ with greedy algorithms (for which $k = 1$), Top-$k$ incurs an additional $k^h$ cost in running time. As mentioned earlier, in practice, we are primarily concerned with fitting small decision trees (e.g., $h = 5$) to the data, as this allows for explainable predictions. In this regard, the additional $k^h$ cost is inexpensive, as confirmed by our experiments.

**The search space of Top-$k$:** We state and prove a simple claim that Top-$k$ returns the *best* tree within its search space.

**Definition 3** (Search space of Top-$k$). *Given a sample $S$ and integers $h, k$, we use $\mathcal{T}_{k,h,S}$ to refer to all trees in the search space of Top-$k$. Specifically, if $h = 0$, this contains all trees with a height of zero (the constant $0$ and constant $1$ trees). For $h \geq 1$, and $\mathcal{I} \subseteq [d]$ being the $k$ coordinates with maximal score, this contains all trees with a root of $x_i$, left subtree in $\mathcal{T}_{k,h-1,S_{x_i=0}}$ and right subtree in $\mathcal{T}_{k,h-1,S_{x_i=1}}$ for some $i \in \mathcal{I}$.*

**Lemma 3.2** (Top-$k$ chooses the most accurate tree in its search space). *For any sample $S$ and integers $h, k$, let $T$ be the output of Top-$k$ with a depth budget of $h$ on $S$. Then,*

$$\mathop{\Pr}_{\boldsymbol{x},\boldsymbol{y}\sim S}[T(\boldsymbol{x}) = \boldsymbol{y}] = \max_{T' \in \mathcal{T}_{k,h,S}}\left(\mathop{\Pr}_{\boldsymbol{x},\boldsymbol{y}\sim S}[T'(\boldsymbol{x}) = \boldsymbol{y}]\right).$$

We refer the reader to Appendix A for the proof of this lemma.

**Parallelizability.** We observe Top-$k$ is largely amenable to parallelization. Observe that each of the $2k$ recursive calls in Step 3 of the pseudocode can be assigned to a separate processor. This means that given $(2k)^h$ processors, a carefully engineered implementation of Top-$k$ can enjoy a parallel running time of $O(hnd)$, which is also the parallel running time of standard greedy algorithms.

## 4 THEORETICAL BOUNDS ON THE POWER OF CHOICES

We refer the reader to the Appendix B for most of the setup and notation. For now, we briefly mention a small amount of notation relevant to this section: We use **bold font** (e.g. $\boldsymbol{x}$) to denote random variables. We also use bold font to indicate *stochastic functions* which output a random variable. For example,

$$\boldsymbol{f}(x) := \begin{cases} x & \text{with probability } \frac{1}{2} \\ -x & \text{with probability } \frac{1}{2} \end{cases}$$

is the stochastic function that returns either the identity or its negation, each with equal probability. To define the data distributions of Theorems 2 and 3, we will give a distribution over the domain, $X$ and the stochastic function that provides the label given an element of the domain.

### 4.1 PROOF OF THEOREM 2

For each depth budget $h$ and search branching factor $K$, we will define a hard distribution $\mathcal{D}_{h,K}$ that is learnable to high accuracy by Top-$K$ with a depth of $h$, but not by Top-$k$ with a depth of $h'$ for any $h' < h + K - k$. This distribution will be over $\{0,1\}^d \times \{0,1\}$, where $d = h + K - 1$. The marginal distribution over $\{0,1\}^d$ is uniform, and the distribution over $\{0,1\}$ conditioned on a setting of the $d$ features is given by the stochastic function $\boldsymbol{f}_{h,K}(x)$. All of the results of this section (Theorems 2 and 3) hold when the feature scoring function is *any* impurity-based heuristic.

**Description of $\boldsymbol{f}_{h,K}(x)$.** Partition $x$ into two sets of variables, $x^{(1)}$ of size $h$ and $x^{(2)}$ of size $K-1$. Let $\boldsymbol{f}_{h,K}(x)$ be the randomized function defined as follows:

$$\boldsymbol{f}_{h,K}(x) = \begin{cases} \mathrm{Par}_h(x^{(1)}) & \text{with probability } 1 - \varepsilon \\ x_i^{(2)} \text{ chosen uniformly at random from } x^{(2)} & \text{with probability } \varepsilon. \end{cases}$$

The definition of $\mathrm{Par}_h(x^{(1)})$ can be found in Appendix B.

The proof Theorem 2 is divided into two components. First, we prove that when the data distribution is $\mathcal{D}_{h,K}$, Top-$K$ succeeds in building a high accuracy tree with a depth budget of $h$. Then, we show that Top-$k$ fails and builds a tree with low accuracy, even given a depth budget of $h + (K - k - 1)$.

**Lemma 4.1** (Top-$K$ succeeds). *The accuracy of Top-$K$ with a depth of $h$ on $\mathcal{D}_{h,K}$ is at least $1 - \varepsilon$.*

**Lemma 4.2** (Top-$k$ fails). *The accuracy of Top-$k$ with a depth of $h'$ on $\mathcal{D}_{h,K}$ is at most $(1/2 + \varepsilon)$ for any $h' < h + K - k$.*

Proofs of both these lemmas are deferred to Appendix B.

### 4.2 PROOF OF THEOREM 3

In this section, we overview the proof Theorem 3, restated for convenience. Some of the proofs are deferred to Appendix B.2

**Theorem 4** (Greediness hierarchy theorem for monotone distributions). *For every $\varepsilon > 0$, depth budget $h$, $K$ between $\Omega(h \log h)$ and $O(h^2/(\log h)^2)$ and $k \leq K - h$, there is a monotone data distribution on which Top-$K$ achieves at least $1 - \varepsilon$ accuracy with a depth budget of $h$, but Top-$k$ achieves at most $0.5 + \varepsilon$ accuracy with a depth budget of $h$.*

Before proving Theorem 4, we'll formalize monotonicity. For simplicity, we'll assume the domain is the Boolean cube, $\{0,1\}^d$, and use the partial ordering $x \preceq x'$ iff $x_i \leq x_i'$ for each $i \in [d]$; however, the below definition easily extends to the domain being any partially ordered set.

**Definition 4** (Monotone). *A stochastic function, $\boldsymbol{f} : \{0,1\}^d \to \{0,1\}$, is* monotone *if, for any $x, x' \in \{0,1\}^d$ where $x \preceq x'$, $\mathbb{E}[\boldsymbol{f}(x)] \leq \mathbb{E}[\boldsymbol{f}(x')]$. A data distribution, $\mathcal{D}$ over $\{0,1\}^d \times \{0,1\}$ is said to be monotone if the corresponding stochastic function, $\boldsymbol{f}(x)$ returning $(\boldsymbol{y} \mid \boldsymbol{x} = x)$ where $(\boldsymbol{x}, \boldsymbol{y}) \sim \mathcal{D}$, is monotone.*

To construct the data distribution of Theorem 4, we will combine monotone functions, Majority and Tribes, commonly used in the analysis of Boolean functions due to their extremal properties. See

Appendix B.2 for their definitions and useful properties. Let $d = h + K - 1$, and the distribution over the domain be uniform over $\{0, 1\}^d$. Given some $x \in \{0, 1\}^d$, we'll use $x^{(1)}$ to refer to the first $h$ coordinates of $x$ and $x^{(2)}$ the other $K - 1$ coordinates. This data distribution is labeled by the stochastic function $\boldsymbol{f}$ given below.

$$\boldsymbol{f}(x) := \begin{cases} \text{Tribes}_h(x^{(1)}) & \text{with probability } 1 - \varepsilon \\ \text{Maj}_{K-1}(x^{(2)}) & \text{with probability } \varepsilon. \end{cases}$$

Clearly $\boldsymbol{f}$ is monotone as it is the mixture of two monotone functions. Throughout this subsection, we'll use $\mathcal{D}_{h,K}$ to refer to the data distribution over $\{0, 1\}^d \times \{0, 1\}$ where to sample $(\boldsymbol{x}, \boldsymbol{y}) \sim \mathcal{D}$, we first draw $\boldsymbol{x} \sim \{0, 1\}^d$ uniformly and then $\boldsymbol{y}$ from $\boldsymbol{f}(\boldsymbol{x})$. The proof of Theorem 4 is a direct consequence of the following two Lemmas, both of which we prove in Appendix B.2

**Lemma 4.3** (Top-$K$ succeeds). *On the data distribution $\mathcal{D}_{h,K}$, Top-$K$ with a depth budget of $h$ achieves at least $1 - \varepsilon$ accuracy.*

**Lemma 4.4** (Top-$k$ fails). *On the data distribution $\mathcal{D}_{h,K}$, Top-$k$ with a depth budget of $h$ achieves at most $\frac{1}{2} + \varepsilon$ accuracy.*

## 5 EXPERIMENTS

**Setup for experiments.** At all places, unless otherwise specified, the Top-1 tree that we compare to is that given by `scikit-learn` (Pedregosa et al., 2011), a standard library for machine learning algorithms. We run experiments on a variety of datasets from the UCI Machine Learning Repository (Dua & Graff, 2017) (numerical as well as categorical features) having a size in the thousands and having $\approx 50 - 300$ features (after binarization). There were $\approx 100$ data sets meeting these criteria, and we took a random subset of 20 such datasets. We binarize all the datasets - for categorical datasets, we convert every categorical feature that can take on (say) $\ell$ values into $\ell$ binary features. For numerical datasets, we sort and compute thresholds for each numerical attribute for an appropriate number of thresholds, so that the total number of binary features is $\approx 100$. A detailed description of the datasets is given in Appendix C.

We build decision trees corresponding to both gini and entropy as the impurity measure $\mathcal{H}$, and report numbers for whichever of the two performed better on the test set (the trends are the same even if we fix the impurity measure). A simple implementation of the Top-$k$ algorithm and other technical details for the experiments will be made publicly available.

### 5.1 KEY EXPERIMENTAL FINDINGS

**Small increments of $k$ yield significant accuracy gains.** Since the search space of Top-$k$ is a superset of that of Top-1 for any $k > 1$, the training accuracy of Top-$k$ is guaranteed to be larger. The primary objective in this experiment is to show that Top-$k$ can outperform Top-1 in terms of test accuracy as well. Figure 2 shows the results for Top-1 versus Top-$k$ for $k = 2, 3, 4$: each plot is a different dataset, where on the x-axis, we plot the depth of the learned decision tree, and on the y-axis, we plot the test accuracy. We also plot the test accuracy of an optimal decision tree (MurTree) in each plot as an additional point of reference.[1] We can clearly observe that the test accuracy increases as $k$ increases—in some cases, the gain is $> 5\%$—and for several datasets, the accuracy of Top-4 is close to that of MurTree.

**Top-$k$ inherits the efficiency and scalability of greedy algorithms.** Since Top-$k$ invests more computation towards fitting a better tree on the training set, its training time is naturally longer than Top-1. However, our experiments show that the slowdown is fairly mild, especially compared to the optimal decision tree algorithm MurTree. As stated in Claim 3.1, the running time of Top-$k$ run to a depth budget of $h$ on a dataset of size $n$ having $d$ binary attributes is $O((2k)^h \cdot nd)$. Thus, if we were to plot the running time of Top-$k$ in log-scale as a function of the depth $h$, we would expect to see a linear behaviour with slope $\log(2k)$. Concretely, Top-1 should differ from Top-$k$ only in its

---

[1]This number should be agnostic to what algorithm is computing the optimal tree, and hence we only compute it for MurTree, since it scales up much better than GOSDT (furthermore, the GOSDT tree is not exactly optimal unless the regularization coefficient is set to 0).

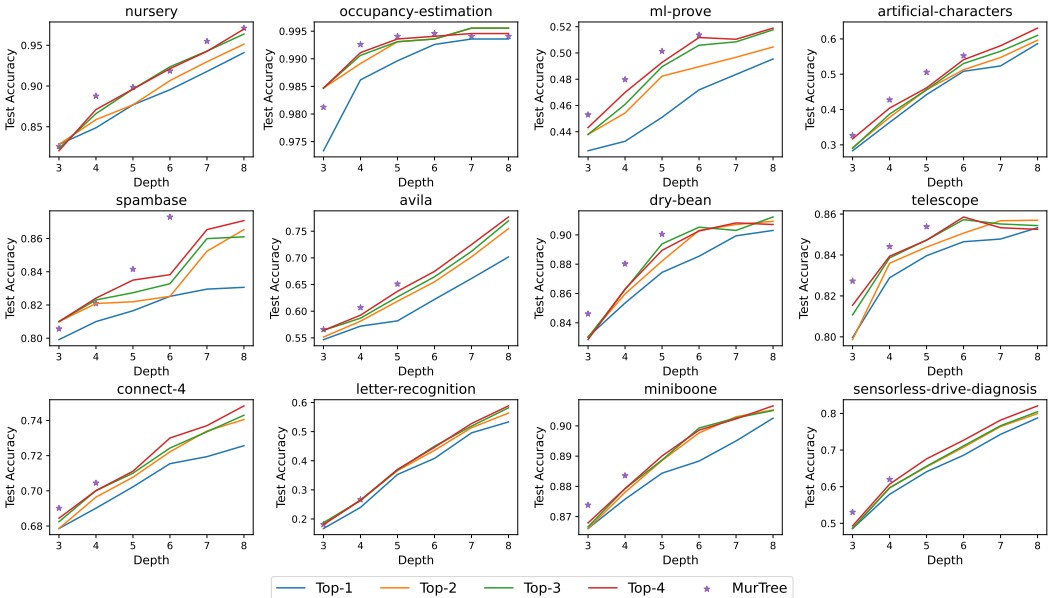

Figure 2: Test accuracy comparison between Top-1, Top-$k$ and MurTree. We can see that Top-$k + 1$ generally obtains higher accuracy than Top-$k$ for $k = 1, 2, 3$, and in some cases, Top-4's accuracy is even comparable to MurTree's. As the dataset size increases, the missing points for MurTree indicate a segmentation fault (possibly due to excessive memory requirements) before the tree-building process was completed.

slope, but the behaviour for both should be linear. Figure 4 in Appendix D confirms this expected behavior. Also, we can very well see that MurTree could not build a decision tree of depth 5 and beyond on the larger datasets, indicating that optimal decision trees do not scale up well (more on this ahead).

Finally, note that we plot two lines for Top-1: Top-1 (sklearn), which is given by `scikit-learn` (and has much better training time) and also Top-1 (basic), where we simply substitute $k = 1$ in our implementation of Top-$k$ (which has comparatively slower training time; in fact, sklearn is $\approx 20\text{x}$ faster). This is simply to highlight the fact that our implementation of Top-$k$ can be made much more efficient by subjecting it to all the optimizations present in `scikit-learn`'s implementation of ID3. We also verified that the accuracies (training and test) for Top-1 (sklearn) and Top-1 (basic) match, so they are building the same trees.

**Top-$k$ scales much better than optimal decision tree algorithms.** Despite having an optimality certificate, optimal decision tree algorithms suffer a lot in running time compared to ID3. Here, we empirically demonstrate that in comparison, Top-$k$ suffers a significantly more benign blow-up in training time. The experiment is identical to that in Figures 14, 15 in the GOSDT paper (Lin et al., 2020), where two notions of scalability are considered. In the first experiment, we fix the number of samples, and gradually increase the number of features to train the decision tree. In the second experiment, we include all the features, but gradually increase the number of samples to train on. The dataset under consideration is the FICO (FICO et al., 2018) dataset, which has a total of 1000 samples having 1407 binary features. On the x-axis, we plot the number of features/samples, and on the y-axis, we plot the training time (in seconds) taken by optimal tree algorithms (MurTree, GOSDT) and Top-$k$. We do this for depth $= 4, 5, 6$ (for GOSDT, the regularization coefficient $\lambda$ is set to $2^{-\text{depth}}$). Figure 3 has the results - we can observe that the training time for both MurTree and GOSDT blows up in a much more drastic manner as compared to Top-$k$, in both the experiments. In particular, for depth $= 5$, both MurTree and GOSDT couldn't finish building a tree on 300 features within the time limit of 10 minutes, while Top-4 completed execution even with all the 1407 features. Similarly, in the latter experiment, GOSDT/MurTree couldn't build a depth-5 tree on 150 samples within the time limit, while Top-4 comfortably finished execution even on 1000 samples. This

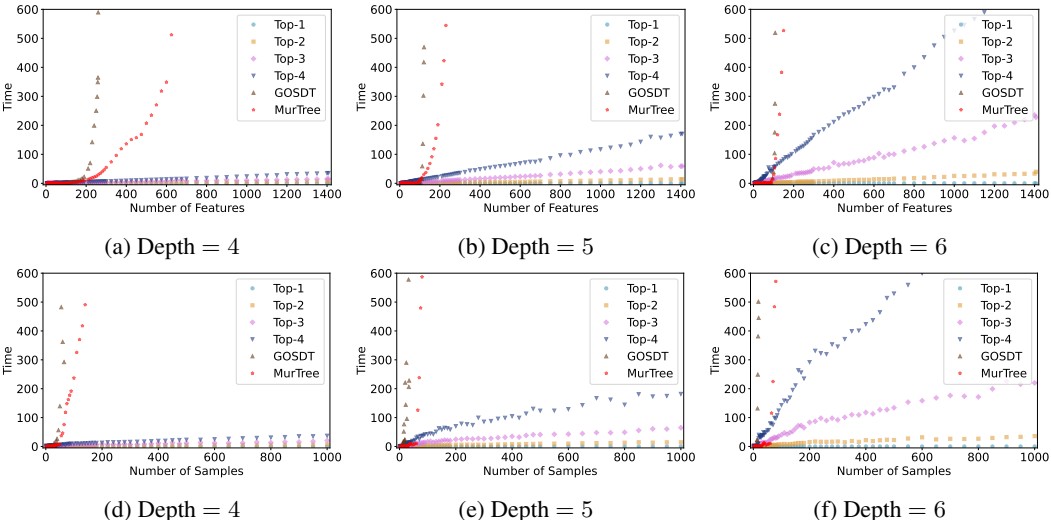

Figure 3: Training time comparison between Top-$k$ and optimal tree algorithms. As the nummber of features/samples increases, both GOSDT and MurTree scale poorly compared to Top-$k$, and beyond a threshold, do not complete execution within the time limit.

nicely illustrates the scalability issues with optimal tree algorithms. Combined with the accuracy gains seen in the previous experiment, Top-$k$ can thus be seen as achieving a nice bargain in the tradeoff between training time and accuracy.[2]

## 6 CONCLUSION

We have shown how popular and empirically successful greedy decision tree learning algorithms can be improved with *the power of choices*: our generalization, Top-$k$, considers the $k$ best features as candidate splits instead of just the single best one. As our theoretical and empirical results demonstrate, this simple generalization is extremely powerful, enabling significant accuracy gains while preserving the efficiency and scalability of standard greedy algorithms. Indeed, we find it surprising that such a simple generalization has not been considered before.

There is much more to be explored and understood, both theoretically and empirically; we list here a few concrete directions that we find particularly exciting and promising. First, we suspect that power of choices affords more advantages over greedy algorithms than just accuracy gains. For example, an avenue for future work is to show that the trees grown by Top-$k$ are more *noise tolerant*. Second, are there principled approaches to the automatic selection of the greediness parameter $k$? Can the optimal choice be inferred from a few examples or learned over time? This opens up the possibility of new connections to machine-learned advice and algorithms with predictions (Mitzenmacher & Vassilvitskii, 2020), an area that has seen a surge of interest in recent years. Finally, as mentioned in the introduction, standard greedy decision tree algorithms are at the very heart of modern tree-based ensemble methods such as XGBoost and random forests. A natural next step is to combine these algorithms with Top-$k$ and further extend the power of choices to these settings.

---

[2]We also ran an experiment comparing accuracy with Soft Decision Trees (Irsoy et al., 2012). We found that their code took significantly longer to train (often 1-2 orders of magnitude) than both Top-3 and Top-4 trained to the same depth. For accuracy, it seems that each method has data sets where it performs better; however, when Soft Decision Trees has more accuracy, it is typically by a very small amount. In contrast, in the other cases, Top-$k$ often has a drastic accuracy gain.

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

## A   PROOFS DEFERRED FROM SECTION 3

*Proof of Lemma 3.2.*   By induction: When $h = 0$, the only trees in the search space are the constant $0$ and constant $1$ functions. Top-$k$ returns which of these two trees is the most accurate.

When $h \geq 1$, let $T'$ be a tree with maximal accuracy within $\mathcal{T}_{k,h,S}$. As $T'$ is in the search space, its root must be one of the $k$ coordinates with maximal score which form the candidate set $\mathcal{I}$.

For each coordinate $i \in \mathcal{I}$, the candidate tree $T_i$ satisfies

$$\Pr_{\boldsymbol{x},\boldsymbol{y} \sim S}[T_i(\boldsymbol{x}) \neq \boldsymbol{y}] = \Pr_{\boldsymbol{x} \sim S}[x_i = 0] \Pr_{\boldsymbol{x},\boldsymbol{y} \sim S}[T_{i0}(\boldsymbol{x}) \neq \boldsymbol{y}] + \Pr_{\boldsymbol{x} \sim S}[x_i = 1] \Pr_{\boldsymbol{x},\boldsymbol{y} \sim S}[T_{i1}(\boldsymbol{x}) \neq \boldsymbol{y}],$$

where $T_{i0}$ and $T_{i1}$ are the left and right subtrees of $T_i$ respectively. Each of $T_{i0}$ and $T_{i1}$ is an output of Top-$k$ with a depth budget of $h - 1$. We assume as the inductive hypothesis that each of these trees minimizes error among all trees in $\mathcal{T}_{k,h-1,S_{x_i=0}}$ and $\mathcal{T}_{k,h-1,S_{x_i=1}}$ respectively; therefore the candidate $T_i$ minimizes error among all trees in $\mathcal{T}_{k,h,S}$ that have $x_i$ at the root. Since Top-$k$ chooses the most accurate of the $T_i$'s, it follows that the chosen tree minimizes error among all trees in $\mathcal{T}_{k,h,S}$. $\qquad\square$

## B   PROOFS DEFERRED FROM SECTION 4

**Setup and notation:**   We use $\mathbb{1}[\cdot]$ for the indicator function, and $[d]$ to refer to the set $\{1, \ldots, d\}$.

For brevity, we'll make two simplifying assumptions about Top-$k$:

1. We will assume Top-$k$ builds *non-redundant* trees, meaning on every root-to-leaf path, each coordinate is queried at most once. This is easy to enforce in the pseudocode: At each step, the algorithm can track a set $Q$ of the coordinates already queried along this path, and pick the top-$k$ coordinates according to the feature score function among $[d] \setminus Q$. For brevity, we do not include that modification to the pseudocode in Figure 1.

2. We assume, roughly speaking, that Top-$k$ has access to an infinite-sized sample. More precisely, whenever Top-$k$ needs to compute an expectation over its sample (to determine the set of $k$ coordinates maximizing the scoring function, or to decide which constant function to put at a leaf), we replace the the empirical expectation with the population expectation. Using standard techniques, if the sample is large enough, these expectations will concentrate and our results still hold.

3. We assume that Top-$k$ always build *complete* trees (i.e every root-to-leaf path has depth exactly $h$). This is without loss of generality, as whenever Top-$k$ stops early, it does so because it has already achieved perfect accuracy on that path.

Throughout this section, we will assume the feature scoring function is an impurity-based heuristic. As our data distribution is uniform on the input, we are able to use the following fact and simultaneously prove results for all impurity-based heuristic.

**Fact B.1** (Proposition 7.7 of Blanc et al. (2020b)). *If the scoring function is* any *impurity-based heuristic, and the data distribution is uniform over inputs ($\boldsymbol{x}$ is uniform when $(\boldsymbol{x}, \boldsymbol{y}) \sim \mathcal{D}$), then the score of a coordinate $i$ is monotone increasing with its correlation with the label, $\mathbb{E}_{(\boldsymbol{x},\boldsymbol{y}) \sim \mathcal{D}}[x_i \boldsymbol{y}]$.*

Fact B.1 means that, when analyzing Top-$k$ on uniform data distributions, we are free to replace the "$k$ coordinates with largest scores" with the "$k$ coordinates with largest correlations."

### B.1   PROOFS DEFERRED FROM SECTION 4.1

The stochastic function $\boldsymbol{f}_{h,K}$ used throughout Lemma 4.1 and Lemma 4.2 combines a function that outputs a random one of $k$ features with the $h$-wise parity function.

**Definition 5** (Parity). *The* parity *function of $\ell$ variables, indicated by $\mathrm{Par}_\ell : \{0,1\}^\ell \to \{0,1\}$, returns*

$$\mathrm{Par}_\ell(x) := \left( \sum_{i \in [\ell]} x_i \right) \mod 2.$$

**Fact B.2** (Computing any function with a complete tree). *Let $f : \{0,1\}^d \to \{0,1\}$ be any function that only depends on the first $h$ variables, meaning there is some $g : \{0,1\}^h \to \{0,1\}$ such that:*

$$f(x) = g(x_{[1:h]})$$

*for all $x \in \{0,1\}^d$. Let $T$ be any non-redundant complete tree of depth-$h$ in which every internal node is one of the first $h$ coordinates. Then, there is a way to label the leaves of $T$ such that $T$ exactly computes $f$.*

*Proof.* Since $T$ is non-redundant, each coordinate is queried at most once on each root-to-leaf path. $T$ is complete and depth-$h$, so each of the first $h$ coordinates must be queried *exactly* once on each root-to-leaf path. Therefore, each leaf of $T$ corresponds to exactly one way to set the first $k$ coordinates of $x$. If the leaf is labeled by the output of $g$ given those first $k$ coordinates, $T$ will exactly compute $f$. □

*Proof of Lemma 4.1.* The function $\mathrm{Par}_h(x^{(1)})$ is a $(1-\varepsilon)$-approximation to $f$, so it suffices to show that the depth-$h$ tree for $\mathrm{Par}_h(x^{(1)})$ is within the search space of Top-$K$ when run to a depth of $h$.

There are only $K-1$ variables not in $x^{(1)}$, so each set of $K$ candidate variables must contain some variable in $x^{(1)}$. Since Top-$K$ is non-redundant, this must be a variable that hasn't yet been queried higher in the tree. Thus, at every step Top-$K$ will always try a candidate variable that reduces the number of relevant $x^{(1)}$-variables by 1. It follows that the complete nonadaptive tree of depth $h$, containing all the variables of $x^{(1)}$, is within the search space, so by Fact B.2 there is a tree in the search space that computes $\mathrm{Par}_h(x^{(1)})$ exactly. Then the accuracy of the output must be at least the total accuracy of this tree, which is $(1-\varepsilon)$. □

*Proof of Lemma 4.2.* Conditioned on any setting of $< k$ variables, for any variable $x_i$ in $x^{(2)}$, $\mathbb{E}[f(x)x_i] \geq 1/k$. Similarly, for any variable $x_j$ in $x^{(1)}$, $\mathbb{E}[f(x)x_j] = 0$. By Fact B.1, at every node the variables of $x^{(2)}$ that have not yet been queried all rank ahead of the variables of $x^{(1)}$. Thus, if at most $K-k$ variables have already been queried, the remaining $k$ most-correlated candidates will all be from $x^{(2)}$, so no variable in $x^{()}$ will be considered. Thus, at least $K-k$ variables from $x^{(1)}$ will be placed.

Since the depth budget $h'$ is smaller than $h+K-k$ and at least $K-k$ variables from $x^{(2)}$ are placed in every path, no path can contain all of the $h$ variables of $x^{(1)}$. The value of $\mathrm{Par}_h(x^{(1)})$ is 0 with probability 1/2 and 1 with probability 1/2 conditioned on the values of any set of variables smaller than $h$. Therefore, the tree built by Top-$k$ cannot achieve accuracy better than 1/2 on the parity portion of the function (and thus have accuracy better than $(1/2 + \varepsilon)$ overall),

□

## B.2 PROOFS DEFERRED FROM SECTION 4.2

The data distribution showing the accuracy separation between Top-$K$ and Top-$k$ is formed by combining the following two functions.

**Definition 6** (Majority). *The* majority *function of $\ell$ variables, indicated by $\mathrm{Maj}_\ell : \{0,1\}^\ell \to \{0,1\}$, returns*

$$\mathrm{Maj}_\ell(x) := \mathbb{1}[\textit{at least half of $x$'s coordinates are 1}].$$

**Definition 7** (Tribes). *For any input length $\ell$, let $w$ be the largest integer such that $(1 - 2^{-w})^{\ell/w} \leq 1/2$. For $x \in \{0,1\}^\ell$, let $x^{(1)}$ be the first $w$ coordinates, $x^{(2)}$, the second $w$, and so on. $\mathrm{Tribes}_\ell$ is defined as*

$$\mathrm{Tribes}_\ell(x) := (x_1^{(1)} \wedge \cdots \wedge x_w^{(1)}) \vee \cdots \vee (x_1^{(t)} \wedge \cdots \wedge x_w^{(t)}) \qquad \textit{where } t := \left\lfloor \frac{\ell}{w} \right\rfloor.$$

For our purposes, it is sufficient to know a few simple properties about Tribes. These are all proven in (O'Donnell, 2014, §4.2).

**Fact B.3** (Properties of Tribes).

1. $\text{Tribes}_\ell$ *is monotone.*

2. $\text{Tribes}_\ell$ *is nearly balanced:*

$$\mathop{\mathbb{E}}_{\boldsymbol{x} \sim \{0,1\}^\ell}[\text{Tribes}_\ell(\boldsymbol{x})] = \frac{1}{2} \pm o(1)$$

   *where the $o(1)$ term goes to $0$ as $\ell$ goes to $\infty$.*

3. *All variables in $\text{Tribes}_\ell$ have small correlation: For each $i \in [\ell]$,*

$$\mathop{\mathbb{E}}_{\boldsymbol{x} \sim \{0,1\}^\ell}[\boldsymbol{x}_i \cdot \text{Tribes}_\ell(\boldsymbol{x})] = O\left(\frac{\log \ell}{\ell}\right).$$

Indeed, the famous KKL inequality implies that any function with the first and second property has a variable with correlation at least $\Omega(\log \ell / \ell)$ (Kahn et al., 1988). Our construction uses Tribes exactly because it has the minimum correlations among functions with the above properties (up to constants). In contrast, we use Maj because its correlations are as *large* as possible, which will "trick" Top-$k$ into building a bad tree.

With the above definitions in-hand, we are able to provide proofs of the two missing Lemmas.

*Proof of Lemma 4.3.* This proof is very similar to that of Lemma 4.1: Once again, we observe the tree computing $(x \mapsto \text{Tribes}_h(x^{(1)}))$ has at least $1-\varepsilon$ accuracy with respect to $\mathcal{D}_{h,K}$. By Lemma 3.2, it is sufficient to prove such a tree is in the search space.

By Fact B.2, any non-redundant complete tree of depth $h$ that only queries the first $h$ coordinates of its input will compute the function $(x \mapsto \text{Tribes}_h(x^{(1)}))$ whenever the leaves are appropriately labeled. Therefore, we only need to prove such a tree is in the search space $\mathcal{T}_{K,h,\mathcal{D}}$. There are only $K - 1$ coordinates that are *not* one of the first $h$ corresponding to $x^{(1)}$. Therefore, within any non-redundant set of $K$ coordinates, at least one must be a non-redundant coordinate from the first $h$. This implies one of the desired trees is in the search space. $\square$

*Proof of Lemma 4.4.* Let $T$ be the tree returned by Top-$k$. Consider any root-to-leaf path of $T$ that does *not* query any of the first $h$ coordinates (those within $x^{(1)}$). Recall that, with probability $(1 - \varepsilon)$, the label is given by $\text{Tribes}_h(x^{(1)})$. On this path, the label of $T$ does not depend on any of the coordinates within $x^{(1)}$. Therefore,

$$\mathop{\Pr}_{(\boldsymbol{x},\boldsymbol{y}) \sim \mathcal{D}_{h,K}}[T(\boldsymbol{x}) = \boldsymbol{y} \mid \boldsymbol{x} \text{ follows this path}]$$

$$= (1 - \varepsilon) \cdot \mathop{\Pr}_{(\boldsymbol{x},\boldsymbol{y}) \sim \mathcal{D}_{h,K}}[T(\boldsymbol{x}) = \text{Tribes}_h(\boldsymbol{x}^{(1)}) \mid \boldsymbol{x} \text{ follows this path}]$$

$$+ \varepsilon \cdot \mathop{\Pr}_{(\boldsymbol{x},\boldsymbol{y}) \sim \mathcal{D}_{h,K}}[T(\boldsymbol{x}) = \text{Maj}_K(\boldsymbol{x}^{(2)}) \mid \boldsymbol{x} \text{ follows this path}]$$

$$\leq (1 - \varepsilon) \cdot \left(\frac{1}{2} + o(1)\right) + \varepsilon \cdot 1 \leq \frac{1 + \varepsilon}{2} + o(1)$$

where the last line follows because $\text{Tribes}_h$ is nearly balanced (Fact B.3). As the distribution over $\boldsymbol{x}$ is uniform, each leaf is equally likely. Therefore, if only $p$-fraction of root-to-leaf paths of $T$ query at least one of the first $h$ coordinates, then,

$$\mathop{\Pr}_{(\boldsymbol{x},\boldsymbol{y}) \sim \mathcal{D}_{h,K}}[T(\boldsymbol{x}) = \boldsymbol{y}] \leq (1 - p) \cdot \left(\frac{1 + \varepsilon}{2} + o(1)\right) + p \cdot 1 \leq \frac{1}{2} + \frac{p}{2} + \frac{\varepsilon}{2} + o(1)$$

Our goal is to prove the tree returned by Top-$k$ achieves at most $\frac{1}{2} + \varepsilon$ accuracy. Therefore, it is enough to prove that $p = o(1)$. Indeed, we will prove that $p \leq O(K^{-2})$.

Here, we apply (Blanc et al., 2020b, Lemma 7.4), which was used to show that Top-1 fails to build a high accuracy tree. They used a different data distribution, but that particular Lemma still applies to

our setting. They prove that a random root-to-leaf path of $T$ satisfies the following with probability at least $1 - O(K^{-2})$: If the length of this path is less than $O(K/\log K)$, at any point along that path, all coordinates within $x^{(2)}$ that have not already been queried have correlation at least $\frac{1}{100\sqrt{k}}$.

That Lemma will be useful for proving Top-$k$ fails with the following parameter choices.

1. By setting $K \geq \Omega(h \log h)$, we can ensure all root-to-leaf paths in $T$ have length at most $O(K/\log K)$, so (Blanc et al., 2020b, Lemma 7.4) applies.

2. By setting $K \leq O(h^2/(\log h)^2)$, we can ensure that all the coordinates within $x^{(1)}$ have correlation less than $\frac{1}{100\sqrt{k}}$ (Fact B.3). This means that all non-redundant coordinates within $x^{(2)}$ have more correlation than those within $x^{(1)}$.

3. By setting $k \leq K - h$, we ensure at all nodes along every path, there are at least $k$ coordinates within the last $K - 1$ coordinates (those corresponding to $x^{(2)}$), that have not already been queried. With probability at least $1 - O(K^{-2})$ over a random path, those all have more correlation than all coordinates within $x^{(1)}$, so Top-$k$ won't query any of the $h$ coordinates within $x^{(1)}$.

We conclude that, with probability at least $1 - O(K^{-2})$ over a random path in $T$, that path does not query any of the first $h$ variables. As a result, the accuracy of $T$ is at most $\frac{1+\varepsilon}{2} + o(1) \leq \frac{1}{2} + \varepsilon$. $\qquad\square$

## C  DETAILS ABOUT DATASETS USED IN SECTION 5

| Name | Type | Size (#train/#test) | #feats | #binary feats | #classes |
|---|---|---|---|---|---|
| connect-4 | C | 67557 (54045/13512) | 42 | 126 | 3 |
| nursery | C | 12960 (10368/2592) | 8 | 27 | 5 |
| letter-recognition | C | 19999 (15999/4000) | 16 | 256 | 26 |
| car | C | 1728 (1382/346) | 6 | 21 | 4 |
| kr-vs-kp | C | 3196 (2556/640) | 36 | 73 | 2 |
| hiv-1-protease | C | 6590 (5272/1318) | 8 | 160 | 2 |
| molecular-biology-splice | C | 3190 (2552/638) | 60 | 287 | 3 |
| mushroom | C | 8124 (6499/1625) | 22 | 117 | 2 |
| artificial-characters | N | 10218 (8174/2044) | 7 | 91 | 10 |
| telescope | N | 19020 (15216/3804) | 10 | 100 | 2 |
| spambase | N | 4601 (3680/921) | 57 | 57 | 2 |
| dry-bean | N | 13611 (10888/2723) | 16 | 96 | 7 |
| occupancy-estimation | N | 10129 (8103/2026) | 16 | 86 | 4 |
| miniboone | N | 130064 (104051/26013) | 50 | 100 | 2 |
| sensorless-drive-diagnosis | N | 58509 (46807/11702) | 48 | 96 | 11 |
| ml-prove | N | 6118 (4588/1530) | 51 | 51 | 6 |
| avila | N | 20867 (10430/10437) | 10 | 100 | 12 |
| taiwanese-bankruptcy | N | 6819 (5455/1364) | 95 | 95 | 2 |
| credit-card | N | 30000 (24000/6000) | 23 | 88 | 2 |
| electrical-grid-stability | N | 10000 (8000/2000) | 13 | 91 | 2 |
| FICO | N | 1000 (900/100) | 23 | 1407 | 2 |

Table 1: Dataset characteristics. In the Type column, C stands for Categorial and N stands for Numerical.

Table 1 provides complete details regarding all the datasets we used in our experiments. For datasets that don't provide an explicit train/test split, we randomly compute a 80:20 split. The column #feats has the number of raw attributes in each dataset, while the column #binary feats has the number of features we obtain after converting these raw attributes to binary-valued attributes. For categorical datasets, we encode a categorical attribute taking on $l$ distinct values to $l$ binary attributes. For numerical datasets, we sort and compute thresholds for each numerical attribute. The number of thresholds is so selected that the total number of binary attributes does not exceed 100.

# D   TRAINING TIME COMPARISON

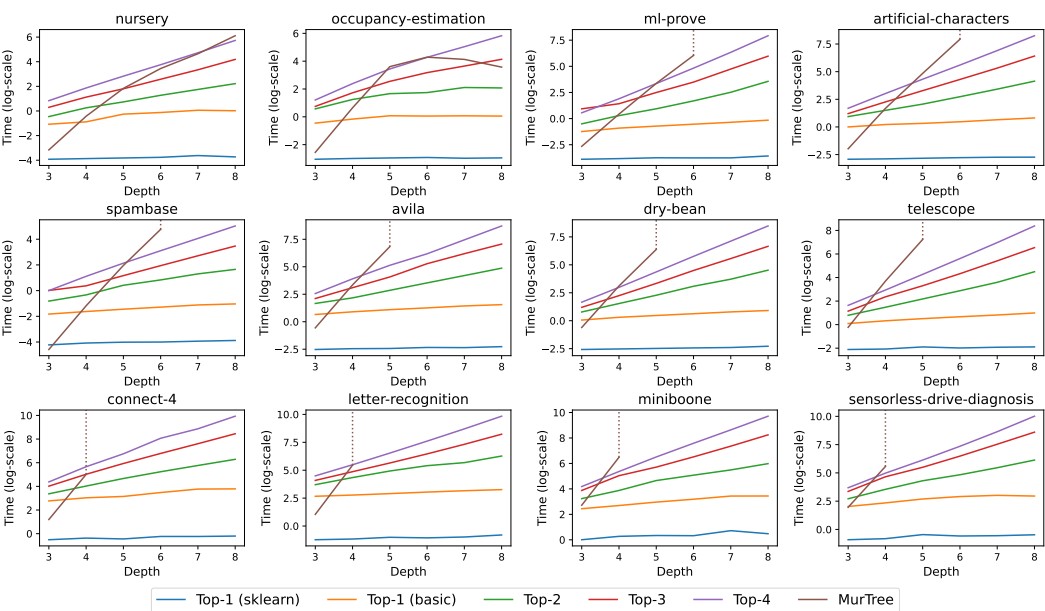

Figure 4: Training time comparison between Top-1, Top-$k$ and MurTree. We can see that the plots for Top-$k$ are all straight lines with increasing slope, as would be expected from Claim 3.1. The dashed vertical line for MurTree indicates a segmentation fault before completing execution of building the decision tree.

# E   ACCURACY COMPARISON WITH TOP-1 - FURTHER PLOTS

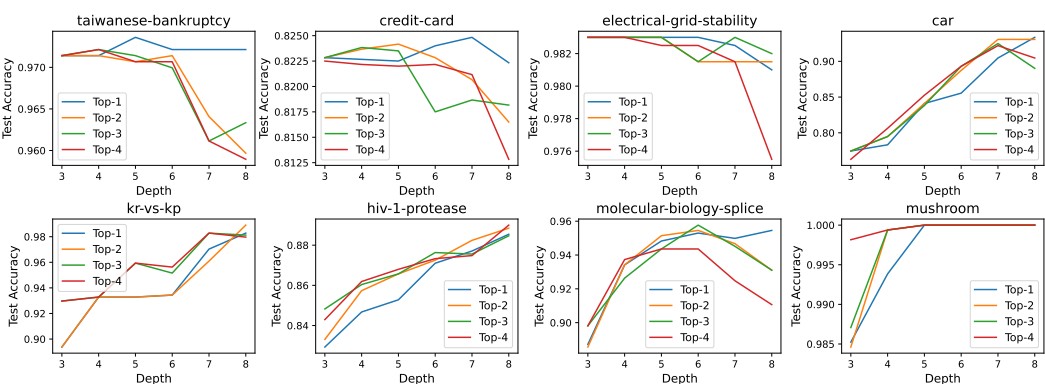

Figure 5: Test accuracy comparison between Top-1 and Top-$k$.

We provide plots from our experiments on a further few datasets comparing the test accuracy of Top-$k$ and Top-1 in Figure 5. In the case of taiwanese-bankruptcy, credit-card and electrical-grid-stability, we can observe that Top-1 is outperforming Top-$k$. However, we believe that this is because the learning problem in this regime is extremely susceptible to overfitting. In particular, we can see that Top-1 is itself not consistently improving with increasing depth. Concretely, increasing depth beyond 3 is already causing Top-1 to overfit, and hence we would expect Top-$k$ to suffer from overfitting even more. Furthermore, we can see that the gradation in the y-axis is very small, in that the accuracy numbers are very close to one another. In the case of the remaining datasets (which all happen to be categorical), while the numbers might not be monotonically getting better with increasing $k$, we can still observe that there is always some value of $k \in \{2, 3, 4\}$ which is outperforming $k = 1$ (except for molecular-biology-splice, for which this is still the case till depth 6). This lends further support to our proposition of incorporating $k$ as an additional hyperparameter to tune while training decision trees greedily.

