# OpenReview forum: "The power of choices in decision tree learning"
_ICLR.cc/2023/Conference — Submitted to ICLR 2023_

### Official Review · Reviewer_Nivx · 2022-10-18

**Confidence:** 4
**Correctness:** 3
**Technical Novelty And Significance:** 3
**Empirical Novelty And Significance:** 4
**Recommendation:** 6

**Clarity, Quality, Novelty And Reproducibility:**

### Clarity and quality
The paper's contributions, goals and empirical analysis are clearly presented. The presentation of the theoretical results should be improved and two intermediate results should be clarified/rectified, as detailed under "Weaknesses - Mathematical rigour".

### Novelty
The proposed method is a straightforward generalization of standard greedy algorithms, so the novelty is somehow limited.

### Reproducibility
The code for reproducing the experiments has not been submitted. However the pseudo-code of the algorithm is reported in Figure 1, and it would be easy to implement the proposed method.

**Strength And Weaknesses:**

## Strengths
### Clarity.
The paper is generally well-written: the contributions are clearly stated, the empirical analysis is described with enough level of detail and intuitions are provided for the theoretical results.

### Significance and impact.
Partly due to its simplicity, the proposed generalization of standard greedy approaches is likely to be adopted in future works, both in the machine learning and the application-oriented communities.

## Weaknesses
### Mathematical rigour.
1. The notation, definitions and theoretical results are split over different sections in the main paper and the appendix. It makes it harder to understand the proofs as the reader has to jump forward and backward just to be able to follow them. More precisely, the theorems are stated in Section 1.1.2, the proofs and definitions are partially reported in Section 4.1 and completed in the appendix. It would really help that at least in the appendix all the elements needed to understand the theorems and the proofs are reported together.

2. Incorrect or unclear results:

- Lemma 3.2. Shouldn't it be a max over the trees and not a min?

- Fact B.2, hence the following results too, stands only for $h \leq k$.

- In proof of Lemma 4.2, the derivation of the two expectations should be reported.

- In proof of Lemma 4.4, why is Pr[T(x) = Tribes(x)| x follows path] is upper bounded by Exp[Tribes(x)|?

3. Missing notation: $x_i$ in Figure 1 (in particular its domain), $x, y, S$ in Definition 2.

### Literature.
The paper never discusses the rich literature on learning optimal soft decision trees with gradient-based methods, e.g. [1], [2]. This literature is very relevant to the proposed work, as it shares the goal of learning decision trees with a scalability close to greedy algorithms and an accuracy close to optimal tree methods.
How does the proposed method compare to this literature?

### Empirical analysis.
1. In page 7, it is said that "We build decision trees corresponding to both gini and entropy as the impurity measure H, and report numbers for whichever of the two performed better on the test set". This is ambiguous and does not seem a rigorous way of comparing methods. Does this mean that within the same figure or subfigure the methods are optimized with different impurity scores? How much the choice of impurity score affect the performance gap?

2. The comparison with the baselines is carried out in terms of accuracy and running times independently. To have a better idea of the trade-off between the two, it would be valuable to plot these two metrics together and highlight in which regimes (in terms of k and depth) it is possible to improve accuracy without incurring into prohibitive running times.

3. Figure 4 is commented in the main paper but reported in the appendix. For the paper's main text to be self-contained, the figure and its description should be reported together.

### (minor) Interpretability.
The theoretical results provide some intuition on how to choose $k$ and $h$, in particular so that learning a top-k tree would allow to reduce the depth of the tree without degrading accuracy. It is not clear however how this analysis relates to the interpretability of the decisions, which is one of the (if not the) principal reasons for learning decision tree-based models. My question is: does using $k > 1$ results in reducing the interpretability of the model, as the overall number of decision splits increases? It would be valuable to briefly discuss this point, showing how this affects the interpretability of the decision for a point and for a class.

### (minor) Typos

- Section 2: understanding the performances -> understanding of the performance

- Section 4 $f_{h, k}$ definition: in the multicase,  $f_{h, k} = $ -> $f_{h, k}(x) = $

- Page 7, footnote: seeing as -> as

- Page 15, last two lines: which was used to that (missing word)

[1] Ozan Irsoy, Olcay Taner Yildiz, Ethem Alpaydin: Soft decision trees. ICPR 2012

[2] Ryutaro Tanno, Kai Arulkumaran, Daniel C. Alexander, Antonio Criminisi, Aditya V. Nori: Adaptive Neural Trees. ICML 2019

**Summary Of The Paper:**

The paper proposes a simple generalization of greedy algorithms for learning decision trees. At a given iteration, instead of growing the tree using the most promising (top-1) feature, the proposed algorithm uses the top-k features and builds a subtree for each of them.
The paper highlights that, even though the time complexity of the top-k algorithm is polynomial in $k$, the algorithm can be easily parallelized to obtain running times of the same order as top-1 algorithms' ones (unlike optimal tree methods).

Unsurprisingly, an empirical analysis shows that using a $k \in [2, 4]$ generally improves over the test accuracy of top-1 greedy algorithms (although it is more prone to overfitting) while having a time complexity (both in number of samples and of features) much smaller than optimal tree methods.
The paper further shows that there exist data distributions on which the test accuracy of the top-K algorithm is provably better than the test accuracy (Theorem 1) when using top-(K-1) for the same depth $h$ or (Theorem 2) when using top-k with $k < K$ and allowing for deeper trees (up to $h + K - k - 1$ depth) or (Theorem 3) when using top-k with $k \leq K - h$ for same depth budget $h$.

**Summary Of The Review:**

The paper's contributions are significant and potentially impactful. The theoretical and empirical analyses are sufficient to support the claims of the paper.
Some work is needed to improve the presentation of the paper, in particular of the theoretical results. I would be happy to increase my score accordingly.

---

> ### Author Response · Authors · 2022-11-11
> **Response to Reviewer Nivx**
>
> We thank the reviewer for their detailed feedback. We are pleased with the reviewer’s summary of the strengths of our paper.
>
> > *"The paper is generally well-written: the contributions are clearly stated, the empirical analysis is described with enough level of detail and intuitions are provided for the theoretical results."*
>
> > *" Partly due to its simplicity, the proposed generalization of standard greedy approaches is likely to be adopted in future works, both in the machine learning and the application-oriented communities."*
>
> ---
>
> Regarding the reviewers concerns,
>
> > *"The theoretical results provide some intuition on how to choose  k  and h, in particular so that learning a top-k tree would allow to reduce the depth of the tree without degrading accuracy. It is not clear however how this analysis relates to the interpretability of the decisions, which is one of the (if not the) principal reasons for learning decision tree-based models. My question is: does using k>1 results in reducing the interpretability of the model, as the overall number of decision splits increases? It would be valuable to briefly discuss this point, showing how this affects the interpretability of the decision for a point and for a class. "*
>
> From the perspective of interpretability, the trained model looks the same regardless of what $k$ is. During training, the algorithm considers more splits, but only one split is used at each node. This is one of the key advantages of Top-$k$. Even though the model is more accurate than Top-1, it is just as easy to interpret.
>
> ---
>
> > *"The paper never discusses the rich literature on learning optimal soft decision trees with gradient-based methods, e.g. [1], [2]. This literature is very relevant to the proposed work, as it shares the goal of learning decision trees with a scalability close to greedy algorithms and an accuracy close to optimal tree methods. How does the proposed method compare to this literature?"*
>
> Thank you for pointing out literature. Following your review, we ran a few preliminary experiments against the code provided by the Soft Decision Trees paper. We found that code took significantly longer to train (often 1-2 orders of magnitude) than both Top-3 and Top-4 trained to the same depth. For accuracy, it seems that each method has data sets where it performs better; however, when Soft Decision Trees has more accuracy, it is typically by a very small amount. In contrast, in the other cases, Top-$k$ often has a drastic accuracy gain over Soft Decision Trees.
>
> We believe that one key advantage of Top-$k$ over these softer decision trees is in interpretability. With Top-$k$, the final tree learned uses extremely simple splits. Furthermore, since the splits are hard (and not soft), to understand the classification of a test point, it is sufficient to look at only one root-to-leaf path. This is easier to understand.
>
> In any case, we will cite and discuss this literature in the next iteration.
>
> ---
>
> > *"The notation, definitions and theoretical results are split over different sections in the main paper and the appendix. It makes it harder to understand the proofs as the reader has to jump forward and backward just to be able to follow them. More precisely, the theorems are stated in Section 1.1.2, the proofs and definitions are partially reported in Section 4.1 and completed in the appendix. It would really help that at least in the appendix all the elements needed to understand the theorems and the proofs are reported together."*
>
> Thank you for this feedback. In the next iteration, we will ensure the appendix is self-contained.
>
> ---
>
> > *"Lemma 3.2. Shouldn't it be a max over the trees and not a min?"*
>
> Typo, thank you for pointing it out.
>
> ---
>
> > *"Fact B.2, hence the following results too, stands only for h≤k"*
>
> We’re not totally sure what the reviewer is referring to here. $k$ does not appear in the statement. Perhaps they mean that it must be the case that $h \leq d$ must hold. This is true, and is enforced based on how we set $d = h + K - 1$ in the Proofs of both Theorem’s 2 and 3 (page 6 and 7 respectively). Feel free to clarify your question if this does not answer it.
>
> ---
>
> > *"In proof of Lemma 4.2, the derivation of the two expectations should be reported."*
>
> We will add this.
>
> ---
>
> >*"In proof of Lemma 4.4, why is Pr[T(x) = Tribes(x)| x follows path] is upper bounded by Exp[Tribes(x)|?"*
>
> We believe the reviewer is referring to the derivation on page 15 in which we say “where the last line follows because Tribes_h is nearly balanced.” At this leaf, the tree will either choose the label 0 or 1. If it chooses 0, the accuracy it achieves on the tribes portion is at most 1 - E[Tribes]. If it chooses 1, the accuracy it achieves is at most E[Tribes]. In both cases, this is at most ½ + o(1).
>
> ---
>
> (character limit reached, to be continued in the next comment)

---

> > ### Author Response · Authors · 2022-11-11
> > **Response to Reviewer Nivx (continued)**
> >
> > > *"In page 7, it is said that "We build decision trees corresponding to both gini and entropy as the impurity measure H, and report numbers for whichever of the two performed better on the test set". This is ambiguous and does not seem a rigorous way of comparing methods. Does this mean that within the same figure or subfigure the methods are optimized with different impurity scores? How much the choice of impurity score affect the performance gap?"*
> >
> > For both Top-1 and Top-k, we used the same approach: Try both impurity measures and report the one with better test accuracy. Due to the consistency, this is a fair way to determine the power of two choices. Regardless, the plots look nearly identical if we just only try one of the impurity functions and will include those in the next iteration.
> >
> > ---
> >
> > > *"The comparison with the baselines is carried out in terms of accuracy and running times independently. To have a better idea of the trade-off between the two, it would be valuable to plot these two metrics together and highlight in which regimes (in terms of k and depth) it is possible to improve accuracy without incurring into prohibitive running times."*
> >
> > We will look into adding plots with both.
> >
> > ---
> >
> > We will address the remaining typos. Thank you for pointing them out!
> >
> > We hope that this clarifies things, and we would be happy to answer any further questions that you have.

---

> > ### Comment · Reviewer_Nivx · 2022-11-15
> > **best tree, revision**
> >
> > I thank the authors for their detailed reply.
> >
> > My questions about interpretability and Fact B.2 (there is no typo there) stemmed from a confusion about the proposed algorithm. I understand now from your answers that only the best tree is retained at the end of training and not all grown ones. I think this point should be made more clear from the introduction. If I didn't miss any reference, the fact that only the tree with maximal score is retained is mentioned only once (in Figure 1).
> >
> > Regarding the comparison with continuous relaxation methods, I cannot find the discussion and results in the paper. In general, I don't think the paper has been revised. If it has been revised, could you provide some pointers?

---

> > > ### Author Response · Authors · 2022-11-15
> > > **Revised submission**
> > >
> > > Thank you again for the detailed review. We have uploaded a revised submission with changes in brown.

---

> > > > ### Author Response · Authors · 2022-11-17
> > > > **Checking in before close of discussion period**
> > > >
> > > > Thank you again for your review! We hope that our response has satisfactorily answered your questions. Please let us know if you have any other questions -- we would be happy to answer them before the discussion period ends.

---

### Official Review · Reviewer_E3aU · 2022-10-25

**Confidence:** 4
**Correctness:** 2
**Technical Novelty And Significance:** 2
**Empirical Novelty And Significance:** 3
**Recommendation:** 3

**Clarity, Quality, Novelty And Reproducibility:**

Except for proofs, the paper is easy to follow. The algorithm seems novel. I think the proofs need a significant revision.

**Strength And Weaknesses:**

Strength
- a simple but effective algorithm for learning more accurate trees than the standard greedy algorithms

Weakness
- The proofs do not seem accurate enough
- implications of the theoretical results

The algorithm is an extension of the standard greedy decision tree algorithms such as ID3. The idea of using top-K features and finding an optimal tree is simple but effective. I agree with the statement that accurate small trees are important and the exponential time complexity is not high when the depth parameter is constant. Empirical results look promising as well.

On the other hand, I have serious concerns about the theoretical results of the paper. First, I think the proofs are not accurate enough since it lacks formal discussion, say, how to obtain information from the distribution. I raised some technical points below.

Second, the implications of the theorems are neither clear nor meaningful. Roughly, the main theorems say that there is a good situation where the algorithm with a large depth parameter is better than that with a smaller depth parameter. This does not imply the larger parameter is always better. Contrarily, when given a small sample, larger trees could overfit and smaller trees are better.



Some technical issues on theoretical proofs
The proofs are not accurate enough in the following reasons.
- In the proofs, the scoring function is not specified. At least, there should be some properties that the function should satisfy in order to make the proofs correct.
- The topK algorithm takes a sample of labeled instances. But, in the theorems and proofs, it somehow has access to the distribution directly(?). You need to formalize the process, e.g., using an oracle. What can the algorithm obtain from the distribution?
- Sample complexity is not discussed. What is the sample complexity of the algorithm? This is related to the previous issue. In general, learning from a larger hypothesis class needs a larger sample, which is well-known in the statistical learning theory (e.g., PAC bound or VC-based generalization bounds).
- The search space \calT need a formal definition. The definition should be defined without using the algorithm.

**Summary Of The Paper:**

The paper considers a simple algorithm for learning decision trees with high accuracy. The authors prove that there exists a distribution over instances for which the algorithm with larger depth parameter k+1 outputs a tree with higher accuracy than that produced by the algorithm with depth parameter k. Furthermore, the author provides experimental results showing the advantages of the proposed algorithm.

**Summary Of The Review:**

The paper proposes a simple but effective decision tree algorithm. Empirical results are promising, showing advantages over previous work. However, theoretical proofs are not accurate enough and I feel the technical contribution is not strong enough unless the theoretical parts are fixed.

---

> ### Author Response · Authors · 2022-11-10
> **Response to Reviewer E3aU**
>
> We thank the reviewer for their feedback and address their concerns below.
>
> ---
>
> > *"Roughly, the main theorems say that there is a good situation where the algorithm with a large depth parameter is better than that with a smaller depth parameter. This does not imply the larger parameter is always better. Contrarily, when given a small sample, larger trees could overfit and smaller trees are better."*
>
> Question for the reviewer: by “depth”, do you mean “choice parameter”? The theorem says that a larger choice parameter can dramatically increase training accuracy (99% vs 51%). Our depth parameter is actually the same for both cases, and therefore the hypothesis class is also the same, and so the same generalization bounds would hold for both.
>
> If the reviewer’s point is that there is some way to draw a training set such that low choice parameter has better test accuracy than high choice parameter, then this true for any two learning algorithms regardless of how they work. For any two learning algorithms A and B that only have access to a training set, there is always some abnormal training set on which A outperforms B and vice versa. In our setting, since both low choice parameter and high choice parameter algorithms return a hypothesis within the same class, it is reasonable to compare training errors. We prove that high choice parameter is always at least as good, and can be dramatically better.
>
> ---
>
> > *"In the proofs, the scoring function is not specified. At least, there should be some properties that the function should satisfy in order to make the proofs correct."*
>
> See Fact B.1: As has been proved in prior work, any scoring function corresponding to a strongly concave impurity function will work for our proofs. This includes the binary entropy function and the Gini index, the two most common choices.
>
> ---
>
> > *"The topK algorithm takes a sample of labeled instances. But, in the theorems and proofs, it somehow has access to the distribution directly(?). You need to formalize the process, e.g., using an oracle. What can the algorithm obtain from the distribution?"*
>
> All our proofs apply in the standard PAC learning framework. Specifically, as long as the sample size is sufficiently large, standard concentration results show that all expectations computed on the sample will be close to those on the whole population, which is sufficient for our lower bounds. We discuss this on page 13 and it is a standard approach (e.g. the literature on statistical query algorithms is based on this).
>
> Formally, our results say the following: We generate a data distribution on which, for a sufficiently large sample, Top-$K$ will have 99% test accuracy but Top-$k$ (for $k < K$) will have 51% test accuracy. A sufficiently large sample is, of course, required for any result of this form.
>
> ---
>
> > *"Sample complexity is not discussed. What is the sample complexity of the algorithm? This is related to the previous issue. In general, learning from a larger hypothesis class needs a larger sample, which is well-known in the statistical learning theory (e.g., PAC bound or VC-based generalization bounds)."*
>
> As clarified above, we are not learning from a larger hypothesis class. We are running Top-$k$ with two different *choice* parameters, but with the same depth budget $h$. The hypothesis class is therefore the same – decision trees of depth $h$ – and the same generalization bounds would hold for both.
>
> ---
>
> We hope that this clarifies things, and we would be happy to answer any further questions that you have.

---

> > ### Author Response · Authors · 2022-11-17
> > **Checking in before close of discussion period**
> >
> > Thank you again for your review! We hope that our response has satisfactorily answered your questions. Please let us know if you have any other questions -- we would be happy to answer them before the discussion period ends.

---

### Official Review · Reviewer_o6dK · 2022-10-25

**Confidence:** 3
**Correctness:** 4
**Technical Novelty And Significance:** 3
**Empirical Novelty And Significance:** 3
**Recommendation:** 8

**Clarity, Quality, Novelty And Reproducibility:**

The paper is clear and well-written. The power of choices, although a known strategy, has been introduced to learning decision trees for the first time. I believe the experiments are simple and can be reproduced.

**Strength And Weaknesses:**

strength: the proposed approach is natural and simple. Authors have shown both theoretical and empirical evidence that shows that their algorithm is better.

weakness: the negative result would have been much stronger had they compared their top-(k+1) algorithm against any other algorithm, not just the top-k algorithm.

**Summary Of The Paper:**

Learning decision trees from data is a simple and rich problem. The classic decision tree learning algorithms use a greedy strategy to select the best feature to evenly split the dataset. In this work, the authors look at k features and select an attribute that achieves maximal accuracy. This recovers the classic strategy for k=1.

They theoretically prove that for certain datasets their top-k algorithm achieves only $1/2+\epsilon$ accuracy whereas top-(k+1) achieves $1-\epsilon$ accuracy.

They also experimentally show that their algorithm betters both the classical top-1 algorithm and also more recent optimization algorithms which uses complicated and non-greedy strategies and hence are time-expensive.

**Summary Of The Review:**

I prefer to accept the paper considering that they combine a classic algorithm with a classic strategy to give an algorithm for learning decision trees.

---

> ### Author Response · Authors · 2022-11-10
> **Response to Reviewer o6dK**
>
> We thank the reviewer for their very positive feedback. We are particularly excited about this comment as it highlights the key conceptual contribution of our work.
>
> > *"The power of choices, although a known strategy, has been introduced to learning decision trees for the first time."*
>
> Indeed, we think that this is an extremely natural idea that could see utility in any context where greedy algorithms are deployed. Our work focuses on the setting of decision tree learning and shows that it indeed leads to theoretical and practical improvements of the state of the art.
>
> Addressing the one weakness the reviewer pointed out:
>
> > *"weakness: the negative result would have been much stronger had they compared their top-(k+1) algorithm against any other algorithm, not just the top-k algorithm."*
>
> In our experiments, we also compare Top-$k$ to the literature on optimal decision trees. In many settings, we show comparable accuracy with drastically reduced runtime.
>
> ---
>
> We hope that this clarifies things, and we would be happy to answer any further questions that you have.

---

> > ### Author Response · Authors · 2022-11-17
> > **Checking in before close of discussion period**
> >
> > Thank you again for your review! We hope that our response has satisfactorily answered your questions. Please let us know if you have any other questions -- we would be happy to answer them before the discussion period ends.

---

### Official Review · Reviewer_uFXj · 2022-10-25

**Confidence:** 4
**Correctness:** 4
**Technical Novelty And Significance:** 3
**Empirical Novelty And Significance:** 3
**Recommendation:** 5

**Clarity, Quality, Novelty And Reproducibility:**

The idea is simple and well-presented. The paper is easy to understand, though there seem to be some duplicated theorem statements and sentences.

**Strength And Weaknesses:**

Strength:

1. The proposed method is a simple and intuitive generalization of the standard tree-fitting algorithms, which naturally covers the optimal tree when $k$ equals the dimension $d$.

2. The new training algorithm is largely amenable to parallelization, despite the exponential increase in computational complexity.

3. The greediness hierarchy theorem demonstrates the power of choices in decision tree learning - even going from $k$ to $k+1$ brings significant improvement in performance.

4. The empirical evaluation also demonstrates the efficacy and efficiency of the proposed algorithm for relatively small $k$.

Questions and weaknesses:

1. One major concern is computational complexity. I'm not fully convinced by the discussion on page 5 about computational complexity. While the original complexity is $2^h$, the proposed algorithm has an additional multiplicative factor of $k^h$. Since this is a multiplicative factor, this is not negligible. The exponential factor is also empirically verified in Fig 4. This is prohibitive for a decent depth - for instance, for h = 10 and k = 3, training one tree in this way is equivalent to training 3^10 ~= 59,000 trees in a greedy way.

2. What is the difference between Theorem 3 and Theorem 4?

3. Scikit-learn is used as a baseline lib for all experiments. However, for tree-fitting algorithms, xgboost or lightgbm seem to be a more common choice and usually deliver better accuracy.

4. In the first paragraph of Section 5, the sentence "For numerical datasets,.... appropriate number of thresholds" is duplicated.

**Summary Of The Paper:**

In this paper, the authors proposed a new way of training decision trees, which extends the standard greedy tree fitting algorithms. The main idea is to consider Top-k features for each split, instead of just greedily using the top-1 feature. Theoretically, the authors present a sharp greediness hierarchy theorem that shows that even a Top-(k+1) tree can be much more power full than Top-k. Further, the empirical results also demonstrate the efficacy and efficiency of the proposed algorithm.

**Summary Of The Review:**

This paper presents a simple and intuitive extension of existing tree-fitting algorithms. One major concern is the multiplicative exponential factor in the computational complexity - I'm not fully convinced this is a scalable solution. See details in the weaknesses.

---

> ### Author Response · Authors · 2022-11-10
> **Response to Reviewer uFXj**
>
> Thank you for your review. We are happy to hear that you like the paper overall. In the following, we address your concerns:
>
> ---
>
> > *"One major concern is computational complexity. I'm not fully convinced by the discussion on page 5 about computational complexity. While the original complexity is , the proposed algorithm has an additional multiplicative factor of . Since this is a multiplicative factor, this is not negligible. The exponential factor is also empirically verified in Fig 4. This is prohibitive for a decent depth - for instance, for h = 10 and k = 3, training one tree in this way is equivalent to training 3^10 ~= 59,000 trees in a greedy way."*
>
> It is also worth comparing us to the significant literature on optimal decision trees (NeurIPS ’19, ICML ‘20, etc.). We achieve comparable accuracy with drastically reduced runtimes. Ultimately, our work gives the practitioner a new, simple to interpret, and powerful parameter. Depending on their computational resources and desired accuracy, they can set it appropriately.
>
> Furthermore, as we mention in our paper (bottom of page 5), and as the reviewer points out themselves,
>
> > *"The new training algorithm is largely amenable to parallelization, despite the exponential increase in computational complexity."*
>
> Given sufficiently many parallel processors, our algorithm has the same run time as Top-1. Ultimately, we agree with the reviewer's comment in spirit but believe it should be framed in a different way: We give power to the practitioner to choose the setting of $k$ most appropriate for them in terms of computational resources.
>
> ---
>
> > *"What is the difference between Theorem 3 and Theorem 4?"*
>
> As Section 4 is dedicated to providing proof sketches of our theorems, we simply restated the statement of Theorem 3 there for convenience of the reader. In the restatement, we make explicit the $\mathrm{polylog}(h)$ dependencies that were being hidden in the $\tilde{O}$ notation in the statement of Theorem 3.
>
> ---
>
> > *"Scikit-learn is used as a baseline lib for all experiments. However, for tree-fitting algorithms, xgboost or lightgbm seem to be a more common choice and usually deliver better accuracy."*
>
> XGBoost and LightGBM are ensemble methods. At each step, they build a decision tree hypothesis by using greedy decision tree algorithms (i.e. what we call Top-1).  Indeed, experiments testing whether combining Top-$k$ with these ensemble methods leads to even more accurate hypotheses is a promising direction for future work, and one we discussed in our conclusion. Since Top-1 is at the very heart of these ensemble methods, we believe that improvements on Top-1 will have downstream implications for these ensemble methods.
>
> It is also worth noting that ensemble methods are significantly less interpretable than a single decision tree. Therefore, in many settings, practitioners stick with a single decision tree hypothesis.
>
> ---
>
> >*"In the first paragraph of Section 5, the sentence "For numerical datasets,.... appropriate number of thresholds" is duplicated."*
>
> This is indeed a typo. Thank you for pointing it out.
>
> ---
>
> We hope that this clarifies things, and we would be happy to answer any further questions that you have.

---

> > ### Author Response · Authors · 2022-11-17
> > **Checking in before close of discussion period**
> >
> > Thank you again for your review! We hope that our response has satisfactorily answered your questions. Please let us know if you have any other questions -- we would be happy to answer them before the discussion period ends.

---

### Decision · Program_Chairs · 2023-01-20

**Decision:**

Reject

**Justification For Why Not Higher Score:**

There is a lack of perspective / zoomed-out reflection of the author on the algorithm provided.
There had been unclear statements

**Justification For Why Not Lower Score:**

The algorithm proposed is sound, and the empirical results are compelling.

**Metareview: Summary, Strengths And Weaknesses:**

 The paper proposes a new way of training decision trees, which extends the standard greedy tree fitting algorithms. The main idea is to consider Top-k features for each split, instead of just greedily using the top-1 feature. A theoretical results is provided that shows that Top-(k+1) can be significantly better than Top-k. Numerical simulations are given, showing the efficacy of the Top-2,3,4 decision trees.

Strengths
- decision trees are key tools for machine learning: having strategies that improve their learning and generalization is key
- the write-up of the paper
- empirical results

Weaknesses
- the authors could state the general optimization problem that decision trees try to solve
- working on the point mentioned just before, if would be nice for the user to call notions that make greedy algorithms to be optimal or almost-optimal (see works on greedy sub modular optimization, on greedy learning on matroid, etc.)
- the incurred computational complexity by the Top-k method could be better described
- there is a lack of clarity in the statement of the results / their proofs.